# Solar geoengineering could redistribute malaria risk in developing countries

Colin J. Carlson [1,2✉], Rita Colwell [3], Mohammad Sharif Hossain [4], Mohammed Mofizur Rahman[5], Alan Robock [6], Sadie J. Ryan [7,8,9], Mohammad Shafiul Alam[4] & Christopher H. Trisos [10,11✉]

Solar geoengineering is often framed as a stopgap measure to decrease the magnitude, impacts, and injustice of climate change. However, the benefits or costs of geoengineering for human health are largely unknown. We project how geoengineering could impact malaria risk by comparing current transmission suitability and populations-at-risk under moderate and high greenhouse gas emissions scenarios (Representative Concentration Pathways 4.5 and 8.5) with and without geoengineering. We show that if geoengineering deployment cools the tropics, it could help protect high elevation populations in eastern Africa from malaria encroachment, but could increase transmission in lowland sub-Saharan Africa and southern Asia. Compared to extreme warming, we find that by 2070, geoengineering would nullify a projected reduction of nearly one billion people at risk of malaria. Our results indicate that geoengineering strategies designed to offset warming are not guaranteed to unilaterally improve health outcomes, and could produce regional trade-offs among Global South countries that are often excluded from geoengineering conversations.

[1] Department of Microbiology and Immunology, Georgetown University Medical Center, Washington, DC 20057, USA. [2] Center for Global Health Science and Security, Georgetown University Medical Center, Washington, DC 20057, USA. [3] University of Maryland, College Park, College Park, MD, USA. [4] Infectious Diseases Division, International Centre for Diarrhoeal Disease Research, Bangladesh (icddr,b), Dhaka, Bangladesh. [5] Institute for Technology and Resources Management in the Tropics and Subtropics, Cologne University of Applied Sciences, Cologne, Germany. [6] Department of Environmental Sciences, Rutgers University, New Brunswick, NJ, USA. [7] Quantitative Disease Ecology and Conservation (QDEC) Lab Group, Department of Geography, University of Florida, Gainesville, FL 32610, USA. [8] Emerging Pathogens Institute, University of Florida, Gainesville, FL 32610, USA. [9] School of Life Sciences, University of KwaZulu-Natal, Durban 4041, South Africa. [10] African Climate and Development Initiative, University of Cape Town, Cape Town, South Africa. [11] Centre for Statistics in Ecology, the Environment and Conservation, University of Cape Town, Cape Town, South Africa. ✉email: colin.carlson@georgetown.edu; christopher.trisos@uct.ac.za

The impacts of climate change are often felt most acutely and personally through human health. Climate change threatens hard-earned progress in infectious disease control and population health during the 20th and early 21st centuries, and most countries' public health infrastructures are underprepared. This is particularly true in developing countries, which already face some of the most severe impacts and the greatest adaptation challenges, often with the least adequate multilateral support. The potential for mass mortality attributable to climate change is increasingly clear[1] and, with existing greenhouse gas emissions mitigation pledges insufficient to keep global warming below 2 °C[2], radical responses to climate change risks are being considered.

Solar geoengineering (also called solar radiation modification, or SRM) is a radical proposal that aims to offset the greenhouse gas-induced climate change by reflecting more sunlight. A wide variety of proposed interventions can be labeled as SRM. These vary in scale of deployment, but all attempt to moderate warming by increasing the amount of sunlight the land, ocean, or atmosphere reflects back to space (e.g., by increasing the reflectivity of clouds over the ocean[3]). The main proposed approach for solar geoengineering is stratospheric aerosol injection (SAI), whereby the deliberate injection of small reflective aerosols into the stratosphere could increase the reflection of incoming sunlight back to space, cooling the planet. SAI has never been tested outdoors, but large volcanic eruptions provide evidence that increasing stratospheric aerosols would cool the planet[4,5]. While responses of temperature and precipitation to SAI have been studied in detail using computer simulations, much less is known about possible consequences for humans or ecosystems[6,7]. Although there is high confidence in the adverse impacts of global warming —and therefore, motivation to avoid unmitigated warming—the climate of a world with SRM would still differ in notable ways from the current or preindustrial climates. As a result, there is no a priori reason to think that solar geoengineering would necessarily improve health outcomes uniformly across regions or health burdens. One study has examined the potential impact of SAI on skin cancer and pollution-related illness[8], while another has explored health impacts of urban heat stress[9]. However, very little is known about possible impacts on infectious diseases, which account for a much higher proportion of global mortality (especially in low- and middle-income countries). This constitutes a major research gap[10], especially given increased attention on solar geoengineering as a potential response to climate risks and the critical need to consider risks to health in policy-making on climate change.

Of all the possible infectious diseases to prioritize for health impact assessments, many of the best candidates are vector-borne diseases, given their massive global burden and their well-demonstrated (and readily forecasted) climate linkages[11,12]. Pathogens transmitted by arthropod vectors like mosquitoes or ticks are particularly sensitive to temperature, which determines both their rate of replication in hosts, and the activity and metabolism of their ectothermic (cold-blooded) vectors. Together, these produce a pattern where their transmission responds unimodally to temperature, in a roughly Gaussian response curve[13]. Thanks to recent advances in experimental and modeling approaches, scientists can rapidly evaluate these response functions, and confidently identify thermal optima ($T_{opt}$) and outer limits of transmission ($T_{min}$ and $T_{max}$). Using these parameters, modelers can identify when climatic conditions fall within the boundaries of transmission, and evaluate how the seasonal window of transmission would change in a warming climate[14,15]. In the best-quantified cases, the temperature can even be explicitly linked to the basic rate of reproduction $R_0(T)$, which quantifies the per-infection ability of the pathogen to transmit onwards[13,14].

Projections of transmission periods and $R_0(T)$ under different climate scenarios can be used to infer whether particular conditions would support epidemic or endemic transmission[15–17].

Thanks to these approaches, an emerging body of evidence shows a high confidence link between global climate change and a potential resurgence of vector-borne diseases. By 2070, climate change is expected to increase the global population at risk of the *Aedes* mosquito-borne dengue fever by up to one billion people[15,18], and could plausibly expose a similar or greater number of people to Zika virus if waning population immunity permits another global epidemic[16]. In Africa, deaths from yellow fever could increase by 10–25%, depending on warming trajectories and vaccination program coverage[19]. But none of these viral infections have a remotely comparable burden to malaria (a protozoan parasite). For comparison, dengue fever is the highest-burden mosquito-borne virus:[20] in 2019, the Americas experienced a world-record 3 million reported cases, including at least 1200 reported deaths. In contrast, the global annual case total of malaria exceeds 200 million, with at least 400,000 deaths almost all caused by the parasite *Plasmodium falciparum*. Malaria remains the sixth largest cause of death in low income countries; though highly effective control efforts have reduced *falciparum* malaria prevalence in Africa by half since the turn of the century[21]—making elimination plausible within a generation[22]— the impacts of COVID-19 might jeopardize this progress[23,24]. Most available evidence suggests that climate change will favor the expansion of *falciparum* malaria into southern Africa and high-elevation regions of east Africa, and possibly reduce transmission in central Africa and the West African Sahel, though less certainty exists on the second point[17,25–29].

Comparatively less attention has been paid to *Plasmodium vivax* malaria, which is responsible for at most 1% of the total global deaths from malaria, but is responsible for 42% of all malaria cases outside Africa, and is likely underreported globally[30]. In the Americas, *P. vivax* is predominant in most malaria reporting countries, and cases are increasing, including co-infections, according to recent work[31]. The collapse of Venezuela's economy, and resulting migration of populations in its wake, following a rapid rise in malaria cases reported from Venezuela, have led to resurgences of both strains of malaria in elimination regions in Latin America[32]. This signals that suitable conditions can give rise to re-seeding and re-establishment of transmission in the absence of strict surveillance and control programs (perhaps a bellwether for the importance of climate shifts). The prospect of encroachment of *vivax* malaria into new areas is particularly troubling, as elimination strategies have largely focused on *P. falciparum*, and basic surveillance may therefore overlook the need to anticipate *P. vivax*, or may simply not have the required testing capacity. Despite these looming risks, *vivax* malaria is often under-represented in climate work, as comparatively less has been known about its ecology.

Two aspects of malaria transmission make this disease a high priority for risk assessment in solar geoengineering research. First, malaria's burden is measurable enough that it impacts economic growth and population-level mortality; if geoengineering is intended to reduce the health risks of climate change in developing countries, perhaps one of the greatest reductions could be experienced through impacts on malaria. Second, for a pathogen with a mostly tropical burden at present, malaria is unusually adapted to cooler temperatures, with peak transmission around 25 °C[33]. It has been hypothesized that malaria transmission could be inadvertently facilitated by SRM[10], especially given that SRM may cool tropical latitudes relatively more[34]. Consequently, when compared to global warming without solar geoengineering, a geoengineering deployment might simply maintain the current burden of disease, or—in a worst-case

scenario—inadvertently increase the burden of malaria in developing countries.

We test this idea by investigating future malaria risk in scenarios where SAI is deployed to stabilize average global temperature at 2020 levels against a background of moderate (Representative Concentration Pathway, or RCP, 4.5) or high future emissions (RCP 8.5), and comparing this to global warming for the same emissions scenarios without geoengineering.

## Results

**Predicting malaria risk in geoengineering scenarios.** We explored two models of geoengineering deployment. First, we used the Geoengineering Model Intercomparison Project (GeoMIP) G3 scenario[35], in which $SO_2$ aerosols are injected into the stratosphere at one point on the equator with injection increasing gradually over the 21st century to offset warming from RCP 4.5 and stabilize temperatures at 2020 levels. Second, the Geoengineering Large Ensemble (GLENS) scenario[36] also increases $SO_2$ injection gradually to stabilize temperatures at 2020 levels, but does this to offset warming from RCP 8.5 and injections are done at four locations away from the equator (30ºN, 30ºS, 15ºN and 15ºS) in order to reduce over-cooling of the tropics. We used three ensemble members from the Earth System Model HadGEM2-ES[37,38] for G3 and RCP 4.5 and from CESM1(WACCM)[36,39] for GLENS and RCP 8.5 (see "Methods"). HadGEM2-ES reaches global warming of on average around 2.8 °C by 2070 compared to pre-industrial climate (1850–1900) while CESM1(WACCM) reaches global warming of on average around 3.3 °C by 2070. The global warming level for 2011–2020 was 1.1 °C above 1850–1900 levels[40]. Geoengineering could be deployed to offset either all or some amount of global warming, and experts on climate politics and policy have noted the challenge of countries agreeing on an optimal amount of geoengineering. We selected the G3 and GLENS scenarios because they deploy geoengineering to offset warming from emissions scenarios widely used in climate impact assessments; the climate models implementing them simulated the more realistic creation of a stratospheric aerosol layer, and they differ in their injection strategies in tropical regions where the burden of malaria is highest. Other scenarios were not used due to potential layers of additional impacts on malaria that might require dedicated attention (e.g., the sudden increase in aerosol injection for implementation of SAI in the G4 scenario).

We used these scenarios to predict the transmission boundaries and intensity of *falciparum* and *vivax* malaria in their respective regions, and project their shifting transmission and seasonality over the next half-century (2020–2070). We limited our analyses to Africa, Asia, and Latin America, where malaria is endemic today, as a subset of the total global shift in favorable temperatures to higher latitudes that has been well-documented for tropical vector-borne diseases. While temperatures may become more favorable for malaria transmission at higher latitudes, healthcare systems in the United States, Europe, and Australia are likely to be strong enough to prevent sporadic importations from establishing malaria transmission in these regions. In contrast, developing countries face the greatest burdens of vector-borne disease, both at present and—with the additional strain of climate change on healthcare systems—in the future.

**Transmission intensity.** We used an approach based on the temperature-dependent basic reproduction number $R_0(T)$, which has become popular as the thermal biology of mosquito-borne disease has become better understood[13,33]. Because arthropod vectors are ectothermic, the processes of mosquito life history and pathogen transmission are very sensitive to temperature, following a simple unimodal curve that can be reconstructed from laboratory experiments and natural transmission data. Given the complexities of real-world transmission, factors like mosquito abundance and human interventions are usually simplified out of this approach, leaving behind a scaled (relative) curve for the temperature-dependent component of $R_0$, the basic reproduction number (that is, the per-capita growth rate of disease transmission; for a full explanation and equations, see "Methods"). For this approach, the $R_0(T)$ expression denotes suitability for transmission as a function of temperature, derived from a fully-parameterized, life-history explicit, temperature-dependent transmission model of both the vector and pathogen, and scaled between zero and one.

This approach cannot necessarily predict total incidence, because major factors such as population density, healthcare access, vector control, or elimination progress are not included. However, mapping $R_0(T)$ can be a first-order proxy of transmission suitability, and, by comparing between scenarios, can indicate where the intensity of transmission and the potential resulting burden of malaria would be higher or lower in different pathways, all other factors being equal. We make two such comparisons: first, we compare possible futures for 2070, with and without SRM deployment; second, we compare a future with SRM deployment against the present-day climate.

We found that across all scenarios, the highest intensity transmission in 2070 will remain where malaria is hyperendemic today, particularly sub-Saharan Africa and the Indian subcontinent (Fig. 1). However, we identified several major differences in transmission suitability, which were more pronounced when geoengineering is used to offset warming from RCP 8.5, the high emissions scenario (Fig. 2). Compared to climate change without solar geoengineering, we found that SRM might substantially reduce malaria transmission in the Indian subcontinent and the Sahel. Relative reductions are also strong in high-elevation areas like the Andes, the Ethiopian highlands, and the Rift Valley, which have been conventionally viewed as the most vulnerable to malaria encroachment in a warming climate, and are also some of the only regions with an identifiable signal of anthropogenic climate change in recent malaria resurgences.

However, we also found that in many tropical regions—including the Amazon basin, Indonesia, west and central Africa, and the southeast and Atlantic coasts of Africa—solar geoengineering in either the G3 or GLENS scenario could, by stabilizing temperatures at colder levels, increase the relative suitability for malaria compared to a future without geoengineering (Fig. 2A, B). This is most pronounced in Brazil, Peru, Ecuador, Venezuela, and Mozambique, which would all warm under RCP 8.5 so much that they reach a near-zero average suitability for malaria. This reduction is projected to be reversed by SRM in the GLENS scenario, which returns these countries to nearly-optimal temperatures for malaria by cooling them up to one degree relative to present day temperatures[34].

Finally, we compared 2070 scenarios with geoengineering to malaria suitability in the present day (2020), and found that a future with SRM deployment still looks markedly different from the present day (Fig. 2C, D). In particular, in the G3 scenario, most of Africa and part of southeast Asia experiences much higher suitability for malaria transmission, while reductions are projected in most of Latin America. In contrast, most of the tropics become less suitable for malaria in the GLENS scenario compared to the present day, especially the Indian subcontinent, but the Americas experience some elevated risk, as does a hotspot in Southeast Asia.

**Seasonality and population at risk.** To explore these patterns further, we identified the outer bounds of transmission from the $R_0(T)$ curves (that is, $T_{min}$ and $T_{max}$). For each first year of the

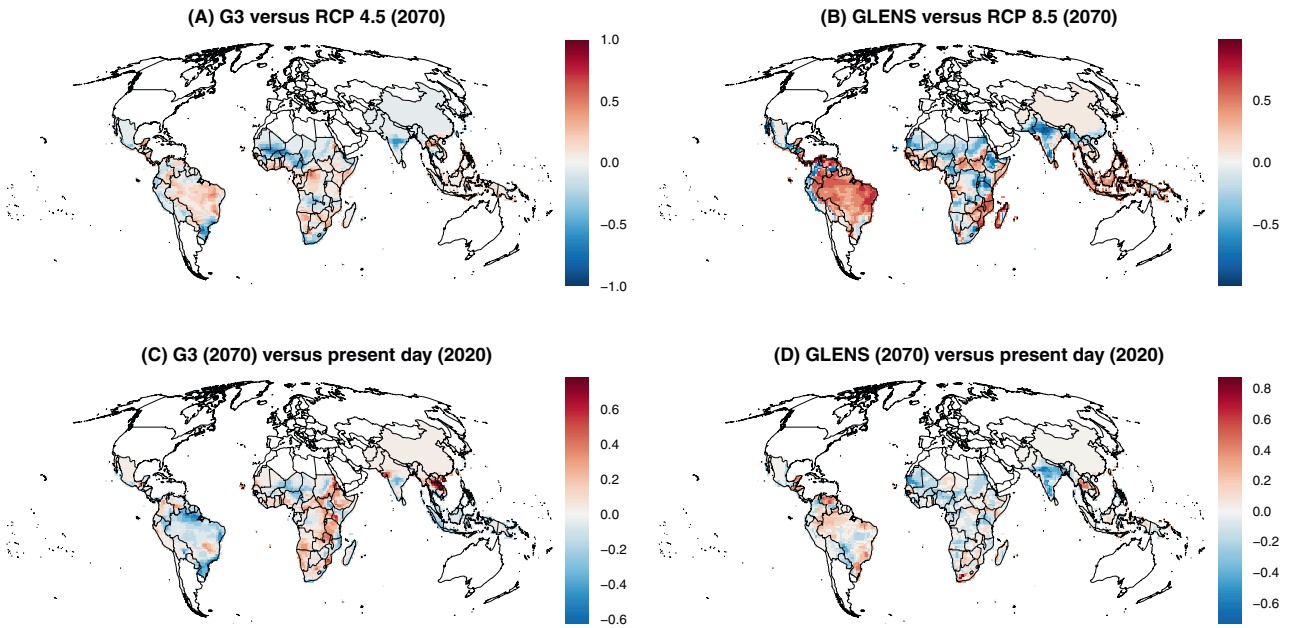

**Fig. 1 Thermal suitability for malaria.** Suitability is measured by a scaled thermal suitability $R_0(T)$, which ranges between zero and one as a function of mean daily temperature, averaged across a full year (every day in 2020 or 2070). Values are mapped for the present day (**A**, **B**), future scenarios without geoengineering (**C**, **D**), and future scenarios with geoengineering (**E**, **F**).

**Fig. 2 Impacts of solar geoengineering on malaria transmission.** Values are given as the difference in scaled thermal suitability $R_0(T)$ between different scenarios, where a higher positive value indicates that the geoengineering scenario creates greater thermal suitability for malaria transmission. Geoengineering scenarios are compared to future scenarios with climate change (**A**, **B**), and compared to the present day (**C**, **D**).

decade, from 2020 to 2070, we evaluated whether each day of the year fit within the bounds of transmission ($T_{min} < T < T_{max}$), and totaled the number of full days that would be conducive to malaria transmission. Using these maps, we split the tropics into regions where malaria transmission risk was unstable (at least one month was conducive for transmission) or stable (at least half the year). Categories similar to these have been previously used to roughly identify where conditions could permit epidemic and endemic transmission, respectively[17,41,42]. Both categories carry risk; today, holoendemic (year-round transmission) malaria places a high burden on children and pregnant women in many hotspots, but episodic malaria epidemics in areas with lower immunity can exhibit high case fatality rates[43]. To approximate these relative burdens, we calculated population at risk from stable and unstable transmission in these areas, using future population projections based on the shared socioeconomic pathways (SSPs) paired with our climate scenarios (SSP2-RCP 4.5 and SSP5-RCP 8.5; see "Methods").

In climate change scenarios without geoengineering, much of the world faces increased malaria risk over at least the next decade; in the longer term, regional redistributions of both *falciparum* and *vivax* malaria risk are expected throughout the tropics, on the order of hundreds of millions of people (Fig. 3 and Supplementary Figs. 1–4). For both RCP 4.5 and RCP 8.5, substantial increases in populations at risk from malaria are projected in east Africa and to a lesser degree central Africa, particularly in high elevation regions where colder temperatures have previously limited malaria transmission. In west Africa, malaria risk also increases, but is dependent on the amount of warming; in RCP 8.5, population exposed to stable transmission risk peaks mid-century and then declines, as much of the region becomes too warm. South and southeast Asia show a similar trend, where warming temperatures in RCP 8.5 lead to massive declines in population at risk (~200 million people in each region), including some populations shifting from stable into unstable risk. In the rest of the tropics, warming temperatures lead to little change or mild declines in total risk, again with shifts from stable to unstable transmission risk.

We find that solar geoengineering to stabilize planetary temperatures at 2020 levels despite moderate greenhouse gas emissions (the G3 scenario) is projected to initially mitigate population at risk of malaria very slightly, but this effect is short-lived and uncertain, and the total population at risk converges on climate change without geoengineering (RCP 4.5) by mid-century (Fig. 3). This largely holds across regions (Supplementary Figs. 1–2); the most pronounced exception to this global pattern (similar risk with and without geoengineering) is in Africa, where cooler temperatures under G3 are projected to lead to large shifts in the population at risk (Supplementary Fig. 1). Specifically, in east Africa, compared to RCP 4.5, geoengineering reduces the expansion of the duration of the transmission season, and thereby leads to a smaller increase in population experiencing stable transmission throughout the century. The opposite pattern is projected for West Africa, where geoengineering is projected to elevate stable transmission risk for roughly 100 million people compared to RCP 4.5.

A larger geoengineering deployment to stabilize planetary temperatures at 2020 levels despite high emissions (the GLENS scenario) would nullify a projected reduction of almost one billion more people at the highest risk of malaria in 2070, compared to climate change without geoengineering (RCP 8.5) (Fig. 3). We also find strong regional patterns for these projected impacts. For *falciparum* malaria, the most striking effect is on west Africa: whereas extreme warming begins to decrease malaria endemicity mid-century, avoiding warming through solar geoengineering allows populations at risk to continue growing, leading to a difference of nearly 200 million people by late in the century (Supplementary Fig. 3). In both south and southeast Asia, RCP 8.5 leads to massive decreases in stable risk (Supplementary Fig. 4). However, averting warming in the GLENS scenario almost entirely nullifies these declines, with a higher population at risk from stable transmission by roughly 200 million people in each region. In east Asia and tropical Latin America, much smaller increases in stable risk are projected due to geoengineering.

Finally, we observed that uncertainty about both baseline climate change impacts and geoengineering impacts showed a tremendous degree of regional variation, which was largely consistent between mid- and high-emissions scenarios. In Latin America and central Africa, changes were highly consistent across climate model simulation runs, and differences between scenarios were minimal. In east and west Africa, we found that differences between climate scenarios were pronounced, but again largely

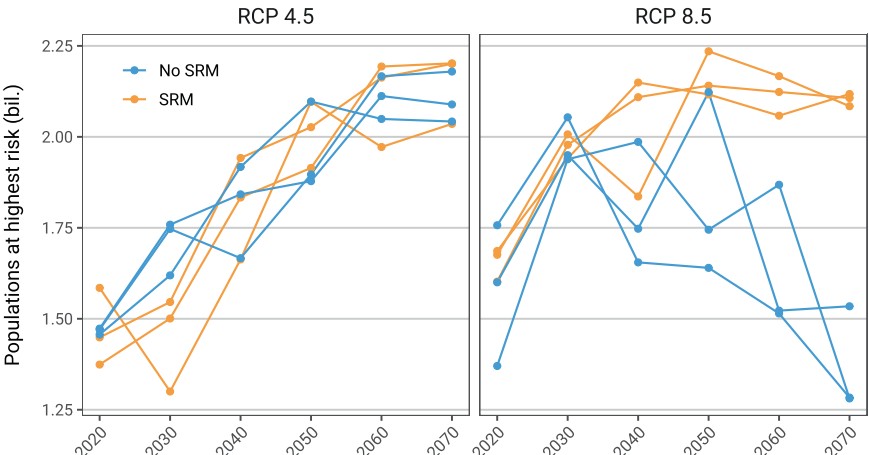

**Fig. 3 Global population at risk from malaria.** Population at highest risk is given as the global total of stable risk (6 months or more of suitability) for both *falciparum* and *vivax* malaria. Scenarios are paired between baseline scenarios and geoengineering scenarios, where solar radiation management deployment is modeled in the G3 scenario relative to Representative Concentration Pathway (RCP) 4.5, and in the Geoengineering Large Ensemble (GLENS) scenario relative to RCP 8.5. Each value is averaged across three scenario-specific runs from a climate model ensemble. (Note that while three scenarios in the two panels follow superficially similar trajectories, the use of different Shared Socio-economic Pathways [SSPs] paired with the RCPs makes results incomparable between RCP 4.5 and RCP 8.5 based scenarios.).

consistent across model runs for a given scenario by 2100. However, we found that southern Asia—a hotspot of projected changes, and the global hotspot of *vivax* malaria endemicity—showed a tremendous degree of variability, so much so that the differences between the RCP 4.5 and G3 scenario are proportionally much smaller, and essentially impossible to evaluate given model uncertainty (Supplementary Fig. 3). This likely reflects a combination of climate model uncertainty and highly aggregated populations, where any one pixel's suitability for malaria may have a marked impact on the total population at risk.

## Discussion

Although the proposed geoengineering scenarios we consider are hypotheticals, our results show that geoengineering may not always have desirable health outcomes. Our simulations indicate that proposed geoengineering schemes could lead to local benefits in east Africa, but also large adverse impacts on west Africa and southern Asia. Totaled across regions, the geoengineering scenarios we examined have at best a neutral but patchwork effect on total malaria risk, and at worst could elevate malaria risk relative to climate change without geoengineering. As such, these scenarios illustrate that solar geoengineering could have adverse impacts on health in cases where the burden of infectious diseases does not strictly increase with warming temperatures. If those diseases are a priority for developing countries' climate policy (as malaria still is in much of the world), we suggest that current proposals for geoengineering might therefore be mismatched to the aim of reducing climate injustice and inequality of climate impacts in the health sector.

For health risks with nonlinear responses to temperature (such as vector-borne diseases), this is likely to be a persistent problem, even for geoengineering deployments that only partially offset global warming. Despite their importance, these health risks are comparatively understudied, in part because climate change impact detection and attribution is more difficult than when evaluating other, more direct health impacts of climate (e.g., heat-related deaths[44]), and projections often follow less intuitive patterns. While reduced warming is likely to have some benefits in places where warming is projected to increase risk, our results show that case by case assessment is urgently needed, and regional differences in outcomes must be disaggregated. Without specific research, assumptions that solar geoengineering's health impacts would be intrinsically fair and effective are unsupported, even compared to the most extreme scenarios for climate change. Emissions reduction for climate change mitigation is widely agreed to produce major net benefits across the health sector; climate geoengineering strategies will likely not be as easily weighed, and may be nearly impossible to optimize across all possible or even all major health burdens, including the hundreds of infectious diseases with poorly-understood, non-linear relationships with warming and other climate variables.

This challenge is particularly pronounced given the difficulty of impact assessment for infectious diseases. For example, in this study, we used a relatively simple approach to measuring population at risk and temperature-sensitive transmission. More advanced models that incorporate human-vector transmission dynamics, and other climate variables like precipitation, have become increasingly accessible;[45] accounting for these variables will only complicate the non-linear relationships between health outcomes and climate geoengineering interventions. Similarly, the relative priority of malaria is dependent not just on climate change scenarios, but also planetary health scenarios[46]. For example, before the COVID-19 pandemic, malaria elimination within a generation was considered plausible, both in terms of epidemiological likelihood and political willpower. After COVID-

19, the burden of malaria—and many other infectious diseases that have been the subject of large-scale elimination programs—hangs in the balance. Even if COVID-19 is eventually contained, most countries' health systems will remain weakened, and some will face a resurgence of diseases that were well-controlled or even nearing elimination before interventions were disrupted[47–49]. In a world that builds back better and recoups these losses, malaria transmission might not be an important issue for climate policy. In a world that fails to do so, with lasting damage from the pandemic, malaria might still be one of the biggest climate-related priorities for developing countries, and therefore, one of the greatest potential downsides (or negative repercussions) of solar geoengineering.

Our study underscores the need for involvement and leadership of developing countries throughout climate change policy, including in geoengineering conversations[50]. This matters particularly for the health sector, where trade-offs may emerge between the priorities of developing countries and of those with the greatest influence in climate policy. Vector-borne diseases are largely projected to shift from the tropics to higher latitudes, especially into more affluent countries with stronger health systems, and that increased risk has dominated conversations about their climate change impacts as compared to trends in the rest of the world. For example, dengue incursion into Europe regularly gathers almost as much attention as the record-setting 2019 dengue epidemics in at least a dozen developing countries. Previous work in this space has cautioned against framing the health impacts of climate change around the encroachment of tropical diseases into less vulnerable Global North countries, but this remains a dominant talking point for climate change impacts on vector-borne disease.

Debate around solar geoengineering could readily fall into similar traps. While geoengineering might prevent or slow the incursion of tropical diseases into higher latitudes, this framing is oblivious to the reality that the global burden of infectious disease falls disproportionately on the same poorer, tropical and subtropical countries (including small island states) that are syndemically vulnerable to future climate impacts and climate injustice. Framing geoengineering around the health impacts of the climate emergency may therefore be undesirable[51], compared to a strategy that meaningfully engages developing countries in terms of their priorities, and develops a comprehensive evidence base of how proposed scenarios will redistribute disease burden between regions. In solar geoengineering research, this means that global involvement in science must not stop with climate modeling; countries' priorities for climate policy, and developing country voices, must be reflected by the impact assessment literature. Only with that greater involvement can the design problem of geoengineering meaningfully reflect its impacts on the people most vulnerable to anthropogenic climate change. At the same time, for the health sector—which has a high stake in global climate politics, but little power over it, especially in the regions most vulnerable to epidemics—these kinds of projections will be an important step towards preparing for an uncertain global future.

## Methods

**Climate projections**. We used two sets of scenarios with different greenhouse gas concentrations and geoengineering deployments. The G3 scenario from the Geoengineering Model Intercomparison Project (GeoMIP)[35] simulates the injection of aerosols into the stratosphere from the equator, starting in 2020 and increasing injection amounts gradually in order to keep global average temperature nearly constant at 2020 levels, despite increasing greenhouse gas concentrations consistent with the moderate mitigation RCP 4.5 scenario. The G3 scenario terminates geoengineering in 2070. Therefore, we compared malaria risk in the G3 scenario to climate change in RCP 4.5 between 2020 and 2070. The Geoengineering Large Ensemble (GLENS)[36] starts injection of stratospheric aerosols at four locations

(30°N, 30°S, 15°N, and 15°S) in 2020 in order to keep average global temperature constant at 2020 levels, despite increasing greenhouse gas concentrations from RCP 8.5, a no mitigation scenario. We compared malaria risk in the GLENS scenario to RCP 8.5 between 2020 and 2070 (Supplementary Figs. 5, 6).

We used projections of near-surface daily air temperature and monthly precipitation from the HadGEM2-ES model[37,38] for both the G3 and RCP 4.5 scenarios and from the CESM1(WACCM) model[36,39] for GLENS and RCP 8.5. We selected these climate models and scenarios in part because, while earlier geoengineering simulations merely turned down the solar constant, both models in this study simulated the more realistic creation of a stratospheric aerosol layer. We note that the selection of these specific climate models' implementation of the scenarios is an unquantified layer of uncertainty in our findings, but similar studies have found that inter-model uncertainty is usually minimal compared to the difference between scenarios with and without SAI[6,52]. Three ensemble members (that is, simulations beginning with slightly different initial conditions) were used from each climate model for each scenario in order to account for natural variability of the climate system, allowing evaluation of the extent to which differences in malaria risk between scenarios with and without geoengineering are forced or random. Because the GLENS scenario is accompanied by only three RCP 8.5 'no geoengineering' runs from CESM1(WACCM), we paired three RCP 8.5 and RCP 4.5 ensemble runs with three ensemble runs generated by each geoengineering simulation, for a balanced study design. All climate variables were regridded to one-degree spatial resolution using bilinear interpolation (Supplementary Figs. 7, 8).

**Malaria transmission models**. Because mosquitoes are ectothermic, mosquito-borne diseases are incredibly temperature-sensitive, and their global distribution and burden is shaped by the underlying landscape of climate. The transmission of mosquito-borne disease is usually depicted as a unimodal response to temperature, with an optimum temperature for transmission bounded above and below by critical limits. When ambient temperature is within those bounds, vectors are considered competent enough to transmit diseases. Transmission is most readily depicted as a relative basic reproduction number $R_0(T)$, which can be rescaled between zero and one for convenience, and is used to identify the minimum, optimum, and maximum temperature for transmission. This can be parameterized with a combination of seven ecological traits (biting rate $a$, vector competence $c$, parasite development rate PDR, mosquito development rate $D$, egg-to-adult mosquito survival $s$, adult mosquito mortality $\mu$, and egg laying per day $E$), using a set of formulas based on the canonical Ross–Macdonald model of malaria transmission. By fitting nonlinear curves to each of the seven traits, a total transmission $R_0(T)$ can be derived:

$$R_0(T) \propto \sqrt{\frac{a^2 csDE \exp(\frac{-\mu}{PDR})}{\mu^3}} \qquad (1)$$

The total value of this formula is then scaled between zero and one. The shape of that curve varies based on each of the component traits, and ultimately differs across different disease transmission systems due to differences in both vector life history and pathogen ecology.

While the $R_0(T)$ approach does not inherently predict the incidence of disease—which is heavily influenced by other factors, like human and vector density, contact rates, and control measures, and also confounded by diagnostics—it does validate well against prevalence data in comparisons. For climate change impact assessment, the $R_0(T)$ approach can be used as a first-principles way to connect warming temperatures to future expansion and redistribution of risk. This approach has been previously used to project the future distribution of risk for malaria[14,17,26], dengue fever[15], and Zika virus[16], among many other vector-borne pathogens.

Basic understanding of thermal ecology is still being developed for malaria. Best understood is *Plasmodium falciparum* malaria, the more severe form of the disease largely transmitted by *Anopheles gambiae* species complex mosquitoes in Africa. A foundational study[33] showed that malaria has a much lower optimal temperature than was originally believed based on observational data, with an optimal temperature around 25 °C. Comparatively less is known about *P. vivax*, the less severe form endemic to southern Asia and the Americas; previously, this pathogen reached much higher latitudes, and has been conventionally viewed as more cold-adapted than *P. falciparum*. More recent work suggests this may be deeply interlinked with vector biology, and that the lower critical temperatures of *P. vivax* may in fact be largely driven by the *An. stephensi* mosquito in south Asia;[53] comparatively less is known about *P. vivax* in the Americas.

To map malaria, we used an approach originally developed over the last decade[26,54], which identifies areas suitable for malaria transmission based primarily on temperature. Following[54], we used daily gridded climate data and mapped the number of days in each pixel that were fully within the thermal bounds of transmission based on $R_0(T)$. Adapting the approach used by previous studies, we masked out deserts based on the most commonly used definition in the geosciences (250 mm annual precipitation or less), based on projected precipitation in each year and scenario. This step accounts for the possibility that some areas are too arid for mosquitoes to complete their reproductive cycle in some areas (though some species, like *An. stephensi*, are container breeders like the yellow fever mosquito *Aedes aegypti*, and may persist in areas with the help of human structures). Beyond this masking step, the presence or absence of *Anopheles* vectors only minimally

limit this approach; roughly a hundred species of *Anopheles* are capable of transmitting malaria on every inhabited continent, and competent vectors occurred throughout the regions we were considering[55]. However, it remains possible that mosquito range shifts—or possible coevolution between mosquitoes and parasites in the future—could change the reliability of this first order proxy, as could the expansion of *P. vivax* and *An. stephensi* mosquitoes in eastern Africa, or any other state shifts in malaria control.

To reflect differences in vector-pathogen systems across landscapes, we used three separate transmission models by region (see below). In sub-Saharan Africa, we used the canonical estimates of *Plasmodium falciparum* transmission by *An. gambiae* (17–34 °C), based on[33]. Several studies since have proposed adjustments to these limits, based on further mathematical modifications of the $R_0(T)$ model (e.g., the inclusion of daily thermal variation through an integral process), but we chose to stay with these estimates as the most similar to the experimental limits that have been observed for transmission (which has been seen as low as 16 °C). For *Plasmodium vivax* in southern Asia, we used a more recent estimate by[53], which assumes that transmission by *An. stephensi* occurs between 15.7 °C and 32.5 °C. For *P. vivax* in the Americas, where at least ten *Anopheles* vectors have been identified as transmitters of malaria[56], as there is little published thermal ecology for individual vector-parasite pairs, we assumed based that local vectors were comparable to *An. gambiae* in their thermal ecology, and that the bounds of *vivax* transmission were 19.4–31.6 °C.

**Projecting populations at risk**. We identified populations at risk by dividing the world into a set of regions modified from the Global Burden of Disease study regions[15,16,57], and estimated the population at risk from falciparum and vivax malaria based on the thermal bounds and precipitation cutoffs explained above. These regions have previously been used for impact assessment models[15,16], but we have made two slight adjustments that primarily reflect the global status of malaria elimination and burden. Given that the Caribbean has eliminated malaria, we reassigned Belize to Latin America (Central), and reassigned Guyana, French Guiana, and Suriname to Latin America (Tropical). Similarly, given the high burden of malaria on the entire island of New Guinea, which is split by the boundary between Indonesia and Papua New Guinea, we regrouped Papua New Guinea with Asia (Southeast).

We paired malaria regions with the dominant form of malaria in each. In Africa, the more severe P. falciparum is the dominant form of malaria, while P. vivax is endemic to South America, southern Asia, and—though rarer than falciparum—parts of east Africa. We therefore modeled falciparum risk in four regions of Africa (West, East, Central, and Southern), and modeled vivax risk in six regions of Asia (East, South, and Southeast) and Latin America (Central, Tropical, and Andean). We focused on falciparum risk for East Africa given both the more severe presentation, and the difficulty of comparing the relative burden of the two given diagnostic challenges.

To estimate future populations at risk, we paired RCPs with shared socioeconomic pathways (SSPs) using the conventional scenario matrix, which identifies more and less plausible combinations of global development strategies and greenhouse gas emissions reductions[58]. Given that RCP and geoengineering scenarios were paired, we chose to keep consistent SSPs between baselines and geoengineering scenarios. In practice, this led to two experiments:[1] RCP 4.5 (some mitigation) paired with SSP2 versus G3 paired with SSP2; and[2] RCP 8.5 (minimal mitigation) paired with SSP5 versus GLENS paired with SSP5. For each climate-SSP pairing we used three climate model runs. SSP data were taken from the SEDAC Global One-Eighth Degree Population Projection Grids[59], which include urban and rural population projections for each decade between 2010 and 2100.

To project populations-at-risk from malaria, we rasterized region boundaries (using a fractional approach to proportional area, rounded to the nearest 1% of a grid cell), and then summed the populations in every grid cell that fell within the minimum and maximum thermal bounds and the precipitation cutoff. Temperature cutoffs were applied at the daily level and summarized for each pixel, and classified into two strata of risk: unstable or epidemic risk (over 30 days and under 180 days, or 1–6 months of the year), and stable or endemic risk (over 6 months of the year), into which different populations were aggregated. These classifications are adapted from prior work that has stratified population at risk into areas of seasonal and endemic risk based on monthly temperature cutoffs[17], but have been adapted to using daily-level climate data. The split can be thought of as indicative of a first-order stratification of burden: most of the burden of malaria, especially mortality, is clustered in places where malaria is endemic or hyperendemic. However, epidemics can be particularly severe in places where malaria is rare, and population immunity is, therefore, lower (e.g., high-elevation communities in east Africa, or near-elimination communities in Latin America).

**Potential impacts of healthcare**. Recent years have seen massive decreases in malaria prevalence in much of the world, especially sub-Saharan Africa, thanks to advances in malaria prevention and treatment. In places where healthcare infra-structure is sufficient to limit disease transmission, or even make advances towards malaria elimination, climate-driven increases in future malaria prevalence may be unlikely, even when population-at-risk or potential transmission intensity has increased. Because our current modeling framework is unable to incorporate the

effect of interventions, or to predict future morbidity and mortality, we are unable to identify these discrepancies directly from model outputs.

However, as a descriptive analysis, we chose to present recent declines in malaria prevalence as a first-order proxy of the effect of interventions (though they may also include the impact of warming temperatures, random interannual variation, or other factors). To do this, we downloaded the most recent (2020) version of the Malaria Atlas Project's global burden estimates for *vivax* and *falciparum* malaria in the years 2000 and 2019 (available from malariaatlas.org/explorer; not yet released in a peer-reviewed study). We mapped these estimates in 2000 and 2019 for *vivax* in Latin America and Asia, and for *falciparum* in Africa, and took the difference as an indication of change over the two-decade interval. Changes over this period include a mix of positive and negative signals, and may be the product of interventions, climatic trends, interannual variation, or other signals (e.g., resurgence after civil conflicts). We mapped these trends against the average $R_0(T)$ in 2020 (averaged across RCP 4.5, RCP 8.5, G3, and GLENS runs), and suggest that a visual comparison of the two may help identify areas where malaria control has been sufficient to decouple transmission potential from actual burden. In areas like these (e.g., Para, Brazil, or Burkina Faso), projected future increases in transmission intensity may not lead to accompanying increases in disease burden, and we encourage caution in the interpretation of our results. These visual comparisons are presented in Supplementary Figs. 9–11.

**Reporting summary**. Further information on research design is available in the Nature Research Reporting Summary linked to this article.

## Data availability

No primary data are generated in this study. Raw climate data used in the analysis are deposited on Dryad at https://doi.org/10.5061/dryad.7m0cfxpwg. Population data, malaria prevalence maps, and thermal suitability curves used in the analysis are available in a Github repository (github.com/cjcarlson/geomalaria), also deposited in Zenodo at https://doi.org/10.5281/zenodo.6054510.

## Code availability

All R scripts needed to reproduce the analysis are available in a Github repository (github.com/cjcarlson/geomalaria), also deposited in Zenodo at https://doi.org/10.5281/zenodo.6054510.

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

## Acknowledgements

This work was supported by funding to MSA from the Solar Radiation Management Governance Initiative (SRMGI) Developing Country Impacts Modeling Analysis for Solar Radiation Management (DECIMALS) grant program. C.H.T. was supported by the FLAIR Fellowship Program: a partnership between the African Academy of Sciences and the Royal Society funded by the UK Government's Global Challenges Research Fund. A.R. was supported by NSF grants AGS-1617844 and AGS-2017113. We thank the UK Hadley Center for contributing their simulations to Earth System Grid Federation, from which we downloaded the output. Special thanks are extended to Simone Tilmes and Daniele Visioni for assistance with climate data, and to Andrew Parker, Peter Irvine, Ben Kravitz, John Shepherd, and John Moore for thoughtful input and comments on the study. icddr,b is grateful to the Government of Bangladesh, Canada, Sweden, and the UK for providing core/unrestricted support.

## Author contributions

C.J.C. and C.H.T. designed experiments and C.J.C. performed analyses. All authors (C.J.C., R.C., M.S.H., M.M.R., A.R., S.J.R., M.S.A., and C.H.T.) contributed to the conceptualization, writing, and editing of the manuscript.

## Competing interests

The authors declare no competing interests.
