## [Peer Review File · Nature Communications]

Reviewer comments, initial review

Reviewer #1 (Remarks to the Author):

This is an interesting and novel piece of work, which simulates the impact of solar radiation management/solar geoengineering on malaria transmission. This work adds solid evidence to the literature around the potential consequences of these novel climate management approaches on human health.

The authors pen a paper that is concise and easy to read and follow. The methods section is detailed in approach and they provide justifications for approaches taken. Overall, I find the paper comprehensive and scientifically sound, with a couple observations:

1. Lines 65-67 and 75-79 needs to be supported by evidence
2. Lines 418-424: It's not clear why the authors decided to use *An. gambiae* thermal limits for *P. vivax* in the Americas, given that *P. vivax* typically transmits at lower temperatures than *P. falciparum*. Also, I am unclear how the authors arrived at the transmission boundaries of 19.4°C to 31.6°C for *P. vivax* in the Americas.

Reviewer #2 (Remarks to the Author):

This study uses current and projected temperature under climate change scenarios to estimate the current and future transmission potential (population at risk) for malaria in Africa, Asia and South America, using empirical estimates of the mosquito's biological and behavioral response to temperature. The study then compares results to those under a modified temperature regime due to the hypothetical use of solar geoengineering.

This is a well written manuscript with a demonstrated understanding of the complexity of climate change and its effects on infectious diseases outcomes. However, I have several issues with the method the authors have used, which I describe below.

Lines: 165:169

“This approach cannot necessarily predict total incidence, because major factors such as population density, malaria control, or elimination progress are not included. However, mapping $R_0(T)$ can be a first-order proxy of transmission suitability, and, by comparing between scenarios, can indicate where the intensity of transmission and the potential resulting burden of malaria would be higher or lower in different pathways. “

The authors used the R_0 approach to quantify risk of malaria transmission, while excluding the effect of population density, and other intervention effects. However

a) it is not clear how these normalized quantities were translated to population at risk Did they use thresholds to classify population into stable and non-stable areas, or were they used to delineate areas of suitability at some threshold values? Not clear);

b) how would they justify the bias in their estimates due to high transmission potential estimates in areas with low human population? Some the highest values of the quantity according to the equation

are sure to be in areas such as the Amazon or remote areas in the Congo where not many people live;

c) The approach does not take in to account the effect of human interventions to control the disease. Given human interventions have led to elimination of the disease in many regions, it may be difficult to take the results seriously if they do not account for human intervention even at the current level with some assumptions going forward. It simply makes it difficult to call for human intervention without accounting for what has already been done. I am afraid, excluding or regrouping countries based on their elimination status may not be enough. One approach the authors could try is using map of prevalence estimates (conveniently covering these regions), which are based on surveys and other sources (Weiss et al 2019), to compare the theoretical R_0 (i.e. your approach after accounting for population density) and these prevalence maps, and assume differences are accounted for by control intervention. This could also help in reporting risk in more interpretable terms (morbidity and to some extent mortality).

463-467

Temperature cutoffs were applied at the daily level and summarized for each pixel, and classified into two strata of risk: unstable or epidemic risk (over 30 days and under 180 days, or 1-6 months of the year), and stable or endemic risk (over 6 months of the year). These classifications are adapted from prior work that has stratified based on monthly temperature cutoffs, but have been adapted to using daily-level climate data.

Not clear how these two strata were used to differentially (if so) estimate population at risk. Also, it would be good to provide citation for previous works.

441:447:

We paired malaria regions with the dominant form of malaria in each. In Africa, the more severe *P. falciparum* is the dominant form of malaria, while *P. vivax* is endemic to South America, southern Asia, and—though rarer than *falciparum*—parts of east Africa. We therefore modeled *falciparum* risk in four regions of Africa (West, East, Central, & Southern), and modeled *vivax* risk in six regions of Asia (East, South, and Southeast) and Latin America (Central, Tropical, and Andean). We focused on *falciparum* risk for East Africa given both the more severe presentation, and the difficulty of comparing the relative burden of the two given diagnostic challenges.

a) Given that most thermal repose curves are based on *P. falciparum* experiments, did the authors use different thresholds for *P. vivax*.

b) Due to the effect relapse after resolution of the primary infection *P. vivax* may present a different challenge to *P. falciparum*. How are the authors factoring this into the risk estimates?

Reviewer #3 (Remarks to the Author):

Review of “Solar geoengineering could redistribute malaria risk in developing countries” by Carlson and others.

This paper investigates the health risks (malaria transmission) of solar geoengineering when implemented on two climate change scenarios, RCP4.5 and RCP8.5. One would expect that this specific health risk would be reduced since solar geoengineering reduces the surface temperatures,

particularly in the tropical regions. However, interestingly, this study finds that the health risk is slightly redistributed in RCP4.5 with solar geoengineering. In the case of RCP8.5 with geoengineering, the risk increases in some regions of the tropics when compared to the scenario without geoengineering. This is not surprising because the science is simple. The transmission rate is not a monotonic function of temperature. The transmission has a window and an optimum temperature (25 deg C). With very warm temperatures (as in the case of RCP8.5), the risk of transmission is reduced as the ambient temperatures are larger than the maximum temperature of the window. Cooling caused by solar geoengineering will, hence, increase the risk by bringing the air temperature back into the window. I believe this is a powerful message that should be told to the public. Basically, this study finds that there are health benefits as well as risks with solar geoengineering. Different regions are affected differently too. The assessment finds that the risk may be increased in some regions even when compared with present-day climate, possibly because of the cooling caused by geoengineering in the tropical regions.

How robust are these results? As this paper has analysed the projections from a single climate model in each scenario, the quantitative results are likely to change with a model change. However, I believe the qualitative result is unlikely to change. The authors should discuss these uncertainties and the robustness aspects in their revision.

The presentation of the material in the paper is sound. I see novelty and originality in the main content, and hence, I recommend acceptance with minor revisions.

Specific comments:

- 1) Line 50: Stratospheric aerosol injection is only one of several proposed solar geoengineering options. The other options may be also briefly mentioned and discussed.
- 2) Line 127: To provide clarity upfront, the authors should discuss here that they are using the outputs from only one climate model (HadGEM2-ES) from the GeoMIP multi-model data. It should be also mentioned here that three ensemble members are used in each case. This will help the readers to quickly understand why there are 3 lines in the figures for each scenario.
- 3) Lines 233-237: The contrast in impacts between the east and west Africa is very interesting indeed. The readers will be able to appreciate the cause for this contrast and other regional differences if the spatial pattern of temperature is shown in the scenarios with and without geoengineering. I suggest the authors to show these temperature plots at least in the supplemental material.
- 4) Lines 354-354: What is the rationale for choosing one climate model from GeoMIP, instead of the multi-model mean?
- 5) Line 363: What is meant by “three natural climate runs”? It is not clear.

Reviewer #1 (Remarks to the Author):

This is an interesting and novel piece of work, which simulates the impact of solar radiation management/solar geoengineering on malaria transmission. This work adds solid evidence to the literature around the potential consequences of these novel climate management approaches on human health.

The authors pen a paper that is concise and easy to read and follow. The methods section is detailed in approach and they provide justifications for approaches taken. Overall, I find the paper comprehensive and scientifically sound, with a couple observations:

1. Lines 65-67 and 75-79 needs to be supported by evidence

We have added references to each of these statements:

“Of all the possible infectious diseases to prioritize for health impact assessments, many of the best candidates are vector-borne diseases, given their massive global burden and their well-demonstrated (and readily forecasted) climate linkages (11,12).”

“In the best-quantified cases, temperature can even be explicitly linked to the basic rate of reproduction $R_0(T)$, which quantifies the per-infection ability of the pathogen to transmit onwards (13,14). Projections of transmission periods and $R_0(T)$ under different climate scenarios can be used to infer whether particular conditions would support epidemic or endemic transmission (15–17).”

2. Lines 418-424: It's not clear why the authors decided to use *An. gambiae* thermal limits for *P. vivax* in the Americas, given that *P. vivax* typically transmits at lower temperatures than *P. falciparum*. Also, I am unclear how the authors arrived at the transmission boundaries of 19.4°C to 31.6°C for *P. vivax* in the Americas.

We appreciate this detailed comment; the temperature limits for transmission of malaria by *Anopheles* mosquitoes was derived using an ecophysiological model first presented in Mordecai *et al.* 2013 *Ecology Letters*, in which components of the mosquito-pathogen life-cycles were explicitly described in their thermal responses, based on laboratory derived measurements of those components at a series of constant temperatures; those were incorporated into a transmission equation for R_0 to determine the shape and limits of the thermal response of malaria transmission. At the time of estimating that first thermal response, there were insufficient laboratory derived data to parameterize separate responses for the two Plasmodia, and the single curve was informed by a combination of parameters from *An gambiae* and related species, infected with either *P. falciparum* or *P. vivax*. In subsequent work (Villena *et al.* 2021 *bioRxiv*), we have refined these thermal response curves to accommodate a distinction between *P. falciparum* and *P. vivax*, but the requisite responses within multiple Anopheline species are lacking; while we have derived these two plasmodial responses in *Anopheles gambiae* and *Anopheles stephensi*, we felt that the latter species was further from representative of species present in the Americas than *An gambiae*. Thus, we are using the *P. vivax* thermal boundaries currently available, and we hope that future research into American vectors will include complete infection experiments across the thermal range to establish an ‘American’ *vivax* thermal response curve.

Reviewer #2 (Remarks to the Author):

This study uses current and projected temperature under climate change scenarios to estimate the current and future transmission potential (population at risk) for malaria in Africa, Asia and South America, using empirical estimates of the mosquito's biological and behavioral response to temperature. The study then compares results to those under a modified temperature regime due to the hypothetical use of solar geoengineering.

This is a well written manuscript with a demonstrated understanding of the complexity of climate change and its effects on infectious diseases outcomes. However, I have several issues with the method the authors have used, which I describe below.

Lines: 165:169

“This approach cannot necessarily predict total incidence, because major factors such as population density, malaria control, or elimination progress are not included. However, mapping $R_0(T)$ can be a first-order proxy of transmission suitability, and, by comparing between scenarios, can indicate where the intensity of transmission and the potential resulting burden of malaria would be higher or lower in different pathways.”

The authors used the R_0 approach to quantify risk of malaria transmission, while excluding the effect of population density, and other intervention effects. However

a) it is not clear how these normalized quantities were translated to population at risk Did they use thresholds to classify population into stable and non-stable areas, or were they used to delineate areas of suitability at some threshold values? Not clear);

The R_0 values are not used to estimate population-at-risk. Instead, the daily thermal cutoffs are applied, which uses the R_0 curve's outer limits but not the actual curve. This is explained in the final paragraph of the methods, now with a brief addition to further clarify:

“To project populations-at-risk from malaria, we rasterized region boundaries (using a fractional approach to proportional area, rounded to the nearest 1% of a grid cell), and then summed the populations in every grid cell that fell within the minimum and maximum thermal bounds and the precipitation cutoff. Temperature cutoffs were applied at the daily level and summarized for each pixel, and classified into two strata of risk: unstable or epidemic risk (over 30 days and under 180 days, or 1-6 months of the year), and stable or endemic risk (over 6 months of the year), into which different populations were aggregated.”

b) how would they justify the bias in their estimates due to high transmission potential estimates in areas with low human population? Some the highest values of the quantity according to the equation are sure to be in areas such as the Amazon or remote areas in the Congo where not many people live;

We appreciate this point, which is why we include the population-at-risk analysis, which overlays areas suitable for malaria transmission with the local gridded population size. This allows us to identify the actual demographic patterns in exposure.

c) The approach does not take in to account the effect of human interventions to control the disease. Given human interventions have led to elimination of the disease in many regions, it may be difficult to take the results seriously if they do not account for human intervention even at the current level with some assumptions going forward. It simply makes it difficult to call for human intervention without accounting

for what has already been done. I am afraid, excluding or regrouping countries based on their elimination status may not be enough. One approach the authors could try is using map of prevalence estimates (conveniently covering these regions), which are based on surveys and other sources (Weiss et al 2019), to compare the theoretical R_0 (i.e. your approach after accounting for population density) and these prevalence maps, and assume differences are accounted for by control intervention. This could also help in reporting risk in more interpretable terms (morbidity and to some extent mortality).

We appreciate this helpful suggestion and have provided the additional analysis suggested by the reviewer as a series of three supplemental figures that present estimated malaria prevalence in 2000, the estimated malaria prevalence in 2019 (the most recent reliable estimates with a published reference), the difference between them as a proxy for the effect of interventions, and the estimated $R_0(T)$ in 2020 (averaged across all scenarios), so that readers can visually compare areas of current transmission, impacts of malaria control, and the underlying thermal propensity for transmission. As noted in the study, a more detailed analysis that actually predicted morbidity and mortality would be impossible without a much more extensive model including socioecological risk factors, healthcare systems, intervention coverage, and vector surveys; however, this is beyond our current scope. Moreover, it is difficult to account for the impact of malaria control to date separate from the impacts of climate warming to date, a topic of extensive debate in the literature; therefore, we are careful to disclaimer these analyses by stressing that they are only a first-order proxy for the effects of interventions. These points are now explained in a new section of the Methods entitled “Potential impacts of healthcare”.

463-467

Temperature cutoffs were applied at the daily level and summarized for each pixel, and classified into two strata of risk: unstable or epidemic risk (over 30 days and under 180 days, or 1-6 months of the year), and stable or endemic risk (over 6 months of the year). These classifications are adapted from prior work that has stratified based on monthly temperature cutoffs, but have been adapted to using daily-level climate data.

Not clear how these two strata were used to differentially (if so) estimate population at risk. Also, it would be good to provide citation for previous works.

A clearer explanation is now provided, including the appropriate reference to previous work:

“To project populations-at-risk from malaria, we rasterized region boundaries (using a fractional approach to proportional area, rounded to the nearest 1% of a grid cell), and then summed the populations in every grid cell that fell within the minimum and maximum thermal bounds and the precipitation cutoff. Temperature cutoffs were applied at the daily level and summarized for each pixel, and classified into two strata of risk: unstable or epidemic risk (over 30 days and under 180 days, or 1-6 months of the year), and stable or endemic risk (over 6 months of the year), into which different populations were aggregated. These classifications are adapted from prior work that has stratified population at risk into areas of seasonal and endemic risk based on monthly temperature cutoffs (17), but have been adapted to using daily-level climate data. The split can be thought of as indicative of a first-order stratification of burden: most of the burden of malaria, especially mortality, is clustered in places where malaria is endemic or hyperendemic. However, epidemics can be particularly severe in places where malaria is rare, and population immunity is therefore lower (e.g., high-elevation communities in east Africa, or near-elimination communities in Latin America).”

441:447:

We paired malaria regions with the dominant form of malaria in each. In Africa, the more severe P. falciparum is the dominant form of malaria, while P. vivax is endemic to South America, southern Asia, and—though rarer than falciparum—parts of east Africa. We therefore modeled falciparum risk in four regions of Africa (West, East, Central, & Southern), and modeled vivax risk in six regions of Asia (East, South, and Southeast) and Latin America (Central, Tropical, and Andean). We focused on falciparum risk for East Africa given both the more severe presentation, and the difficulty of comparing the relative burden of the two given diagnostic challenges.

a) Given that most thermal repose curves are based on *P. falciparum* experiments, did the authors use different thresholds for *P. vivax*.

Yes. The Villena work (2021 *bioRxiv*) establishes the differentiated thermal responses for the two plasmodia. As we explain in the Methods:

“To reflect differences in vector-pathogen systems across landscapes, we used three separate transmission models by region (see below). In sub-Saharan Africa, we used the canonical estimates of *Plasmodium falciparum* transmission by *An. gambiae* (17 to 34°C), based on (33). Several studies since have proposed adjustments to these limits, based on further mathematical modifications of the $R_0(T)$ model (e.g., the inclusion of daily thermal variation through an integral process), but we chose to stay with these estimates as the most similar to the experimental limits that have been observed for transmission (which has been seen as low as 16°C). For *Plasmodium vivax* in southern Asia, we used a more recent estimate by (53), which assumes that transmission by *An. stephensi* occurs between 15.7°C and 32.5°C. For *P. vivax* in the Americas, where at least ten *Anopheles* vectors have been identified as transmitters of malaria (56), as there is little published thermal ecology for individual vector-parasite pairs, we assumed based that local vectors were comparable to *An. gambiae* in their thermal ecology, and that the bounds of *vivax* transmission were 19.4°C to 31.6°C.”

b) Due to the effect relapse after resolution of the primary infection *P. vivax* may present a different challenge to *P. falciparum*. How are the authors factoring this into the risk estimates?

In this study, we are using the ecophysiological model – the thermal response curves – to project the suitability of transmission as a function of temperature; this is different from using case data to derive a model in a phenomenological framework. We do not use human case data, which would pose the challenge of relapsing cases of *P. vivax* generating cases in locations where the transmission event had not occurred, giving rise to false estimates of environmental conditions for transmission. Therefore, this issue is not a key problem for our methodology.

Reviewer #3 (Remarks to the Author):

Review of “Solar geoengineering could redistribute malaria risk in developing countries” by Carlson and others.

This paper investigates the health risks (malaria transmission) of solar geoengineering when implemented on two climate change scenarios, RCP4.5 and RCP8.5. One would expect that this specific health risk would be reduced since solar geoengineering reduces the surface temperatures, particularly in the tropical regions. However, interestingly, this study finds that the health risk is slightly redistributed in RCP4.5

with solar geoengineering. In the case of RCP8.5 with geoengineering, the risk increases in some regions of the tropics when compared to the scenario without geoengineering. This is not surprising because the science is simple. The transmission rate is not a monotonic function of temperature. The transmission has a window and an optimum temperature (25 deg C). With very warm temperatures (as in the case of RCP8.5), the risk of transmission is reduced as the ambient temperatures are larger than the maximum temperature of the window. Cooling caused by solar geoengineering will, hence, increase the risk by bringing the air temperature back into the window. I believe this is a powerful message that should be told to the public. Basically, this study finds that there are health benefits as well as risks with solar geoengineering. Different regions are affected differently too. The assessment finds that the risk may be increased in some regions even when compared with present-day climate, possibly because of the cooling caused by geoengineering in the tropical regions.

How robust are these results? As this paper has analysed the projections from a single climate model in each scenario, the quantitative results are likely to change with a model change. However, I believe the qualitative result is unlikely to change. The authors should discuss these uncertainties and the robustness aspects in their revision.

As the reviewer points out, the major results and their implications are unlikely to change particularly notably between scenarios. However, as requested by the reviewer, we now flag this as a potential point of uncertainty and refer to a pair of studies that have previously examined some of the same scenarios and found minimal uncertainty in this aspect:

“...We selected these climate models and scenarios in part because, while earlier geoengineering simulations merely turned down the solar constant, both models in this study simulated the more “realistic” creation of a stratospheric aerosol layer. We note that the selection of these specific climate models’ implementation of the scenarios is an unquantified layer of uncertainty in our findings, but similar studies have found that inter-model uncertainty is usually minimal compared to the difference between scenarios with and without SAI (6,52).”

The presentation of the material in the paper is sound. I see novelty and originality in the main content, and hence, I recommend acceptance with minor revisions.

Specific comments:

1) Line 50: Stratospheric aerosol injection is only one of several proposed solar geoengineering options. The other options may be also briefly mentioned and discussed.

We have added text to the introduction stating that a wide variety of proposed interventions that increase the amount of sunlight reflected by the land, ocean or atmosphere can be labelled as solar geoengineering and mentioning marine cloud brightening as one example.

2) Line 127: To provide clarity upfront, the authors should discuss here that they are using the outputs from only one climate model (HadGEM2-ES) from the GeoMIP multi-model data. It should be also mentioned here that three ensemble members are used in each case. This will help the readers to quickly understand why there are 3 lines in the figures for each scenario.

We have added text to this methods paragraph in the introduction, stating that we used three ensemble members from HadGEM2-ES for G3 and RCP4.5 and three ensemble members from CESM1(WACCM) for GLENS RCP8.5

3) Lines 233-237: The contrast in impacts between the east and west Africa is very interesting indeed. The readers will be able to appreciate the cause for this contrast and other regional differences if the spatial pattern of temperature is shown in the scenarios with and without geoengineering. I suggest the authors to show these temperature plots at least in the supplemental material.

We appreciate this suggestion and have added a new supporting information document that includes four supplemental figures, which show the mean annual temperature in 2020 and 2070 across all scenarios, as well as the difference between 2070 and 2020 in each.

4) Lines 354-354: What is the rationale for choosing one climate model from GeoMIP, instead of the multi-model mean?

We did not use a multi-model mean because only one model – HadGEM2-ES – simulated a stratospheric aerosol cloud for the GeoMIP G3 scenario and completed multiple ensemble runs, so we could only use this model. IPSL-CM5-LR only completed a single model run for G3. More climate models ran multiple ensemble members for the G4 geoengineering scenario from GeoMIP but we did not use that scenario in this study because G4 has a sudden implementation of aerosol injections at 5 Tg SO₂ from 2020, instead of a gradual increase in injection amounts. This feature is unusual and might have more complicated impacts on health that we felt were beyond the scope of our study, as we note in the introduction.

However, we have also added a note about the uncertainty that might come from model selection: “We note that the selection of these specific climate models’ implementation of the scenarios is an unquantified layer of uncertainty in our findings, but similar studies have found that inter-model uncertainty is usually minimal compared to the difference between scenarios with and without SAI (6,52).”

5) Line 363: What is meant by “three natural climate runs”? It is not clear.

We removed the word ‘natural’ and have revised the text to clarify that we meant the no geoengineering RCP 4.5 and RCP 8.5 ensemble members.

Reviewer comments, second review

Reviewer #3 (Remarks to the Author):

I am overall satisfied with the revision and the responses except for the following:

For my comment on showing the spatial pattern of temperature change in the scenarios with and without geoengineering, the response is "have added a new supporting information document that includes four supplemental figures, which show the mean annual temperature in 2020 and 2070 across all scenarios, as well as the difference between 2070 and 2020 in each." However, I do not find the figures in the revised submission which has 7 extended data figures.

Line 49-50: For consistency with the literature, " Among these proposed schemes is solar geoengineering (also called solar radiation modification, or SRM). A wide variety of proposed interventions can be labelled as SRM." should be changed to "Solar geoengineering (also called solar radiation modification, or SRM) is a radical proposal that aims to offset the greenhouse gas-induced climate change by reflecting more sunlight. A wide variety of options have been proposed under SRM"

Reviewer #4 (Remarks to the Author):

This well-written paper explores the effects of geoengineering on malaria risk. The introduction section is well researched and very clearly explains how this manuscript fits in to ongoing conversations and research in this field.

I was asked by the editor to comment on the authors' response to Reviewer #2, who was not available to comment on this revised draft. I believe that Reviewer #2's comments have been addressed.

The authors added details and clarifications to the manuscript to better explain their methods and address the questions raised by the reviewer. The new section on healthcare impacts adds important context to the study results.

I have a few additional comments:

1. Scaled R0

The choice to scale R0 is odd. R0 has a specific meaning in infectious disease epidemiology – the average number of secondary infections resulting from an index case in an entirely susceptible population. R0 below 1 means transmission is not sustained. R0s of various diseases are known and compared with each other. Scaling the value from 0 to 1 removes all of this context.

Reading through several related prior publications, I see that the justification given in Ryan et al (PTND, 2019) for scaling this value is that two of the parameters in the true R0 denominator (human population size N and human infection recovery rate r) are difficult to obtain at large scales:

[equation did not paste properly, see first equation of methods in <https://journals.plos.org/plosntds/article?id=10.1371/journal.pntd.0007213>]

To bypass this problem, the authors in Ryan 2019 omitted these two parameters and scaled the remainder of the equation, resulting in the proportional relationship presented in line 389 of this manuscript.

However, the difficulty in obtaining N and r is even further reason not to scale the value across locations. The equation in line 389 holds if N and r are constant. But since N and r vary in space (and time), the scaled value on the right hand side of the equation does not have the same relationship to the actual R_0 across locations.

I would recommend giving the scaled quantity a new name and explain that it is the temperature-dependent component of R_0 rather than inaccurately calling it R_0 .

2. Aridity

Regarding the aridity mask (Line 414), is the mask updated in each of the scenarios according to projected precipitation? Please clarify in the text. If the mask is not updated, this limitation should be mentioned as precipitation can be as important in defining the limits of transmission suitability.

I believe both of these comments can be easily addressed by the authors, and am happy to recommend the manuscript for publication.

REVIEWERS' COMMENTS

Reviewer #3 (Remarks to the Author):

I am overall satisfied with the revision and the responses except for the following:

For my comment on showing the spatial pattern of temperature change in the scenarios with and without geoengineering, the response is "have added a new supporting information document that includes four supplemental figures, which show the mean annual temperature in 2020 and 2070 across all scenarios, as well as the difference between 2070 and 2020 in each." However, I do not find the figures in the revised submission which has 7 extended data figures.

We apologize for this error and have fixed this – the supplement was not correctly uploaded in the last round of revision! Everything is now in one master file and should be correctly uploaded.

Line 49-50: For consistency with the literature, " Among these proposed schemes is solar geoengineering (also called solar radiation modification, or SRM). A wide variety of proposed interventions can be labelled as SRM." should be changed to "Solar geoengineering (also called solar radiation modification, or SRM) is a radical proposal that aims to offset the greenhouse gas-induced climate change by reflecting more sunlight. A wide variety of options have been proposed under SRM"

We have made this change as requested.

Reviewer #4 (Remarks to the Author):

This well-written paper explores the effects of geoengineering on malaria risk. The introduction section is well researched and very clearly explains how this manuscript fits in to ongoing conversations and research in this field.

I was asked by the editor to comment on the authors' response to Reviewer #2, who was not available to comment on this revised draft. I believe that Reviewer #2's comments have been addressed.

The authors added details and clarifications to the manuscript to better explain their methods and address the questions raised by the reviewer. The new section on healthcare impacts adds important context to the study results.

We thank the reviewer for the positive feedback.

I have a few additional comments:

1. Scaled R0

The choice to scale R0 is odd. R0 has a specific meaning in infectious disease epidemiology – the average number of secondary infections resulting from an index case in an entirely susceptible population. R0 below 1 means transmission is not sustained. R0s of various

diseases are known and compared with each other. Scaling the value from 0 to 1 removes all of this context.

Reading through several related prior publications, I see that the justification given in Ryan et al (PTND, 2019) for scaling this value is that two of the parameters in the true R_0 denominator (human population size N and human infection recovery rate r) are difficult to obtain at large scales:

[equation did not paste properly, see first equation of methods in <https://journals.plos.org/plosntds/article?id=10.1371/journal.pntd.0007213>]

To bypass this problem, the authors in Ryan 2019 omitted these two parameters and scaled the remainder of the equation, resulting in the proportional relationship presented in line 389 of this manuscript.

However, the difficulty in obtaining N and r is even further reason not to scale the value across locations. The equation in line 389 holds if N and r are constant. But since N and r vary in space (and time), the scaled value on the right hand side of the equation does not have the same relationship to the actual R_0 across locations.

I would recommend giving the scaled quantity a new name and explain that it is the temperature-dependent component of R_0 rather than inaccurately calling it R_0 .

We appreciate the reviewer's point and now refer to this quantity in-text and in the methods as the "temperature-dependent component of the basic reproduction number $R_0(T)$," as well as already explaining that it serves as a "thermal suitability" metric and using that term throughout.

2. Aridity

Regarding the aridity mask (Line 414), is the mask updated in each of the scenarios according to projected precipitation? Please clarify in the text. If the mask is not updated, this limitation should be mentioned as precipitation can be as important in defining the limits of transmission suitability.

We now clarify that the aridity mask is indeed applied "based on projected precipitation in each year and scenario."

I believe both of these comments can be easily addressed by the authors, and am happy to recommend the manuscript for publication.